# Estimation of the Antenna Phase Center Correction Model for the BeiDou-3 MEO Satellites

**Xingyuan Yan [1], Guanwen Huang [1,\*], Qin Zhang [1], Le Wang [1], Zhiwei Qin [1] and Shichao Xie [1]**

College of Geology Engineering and Geomatics, Chang'an University, Xi'an 710054, China;
yanxydice@chd.edu.cn (X.Y.); dczhangq@chd.edu.cn (Q.Z.); wangle18@chd.edu.cn (L.W.);
gnssorbitqzw@chd.edu.cn (Z.Q.); wonderwall@chd.edu.cn (S.X.)

**\*** Correspondence: huang830928@chd.edu.cn; Tel.: +86-029-82339043

**Abstract:** Satellite antenna phase center offsets (PCOs) and phase variations (PVs) for BeiDou-3 satellites are estimated based on the tracking data of the Multi-GNSS Experiment (MGEX) and the international GNSS Monitoring and Assessment System (iGMAS) network. However, when estimating the (PCOs) of BeiDou-3 medium Earth orbit (MEO) satellites by pure Extending the CODE Orbit Model (ECOM1), the x-offset estimations of the PCOs have a systematic variation of about 0.4 m with the elevation of the Sun above the orbital plane (β-angle). Thus, a priori box-wing solar radiation pressure (SRP) model of BeiDou-3 MEO was assisted with ECOM1. Then, the satellite type-specific PCOs and common PVs were obtained. The estimations of PCOs and PVs were compared with the MGEX PCOs from the precise orbit and clock offset. When the MGEX PCOs were used, the root mean square (RMS) of 24 h overlap was 6.76, 4.36, 1.46 cm, in along-track, cross-track, and radial directions, respectively; the RMS and standard deviations (STD) of the 24 h clock offset overlap were 0.28 and 0.15 ns; the fitting RMS of the 72 h clock offset of the quadratic polynomial was 0.243 ns. After comparing this with the estimated PCOs and PVs, the RMS of the 24 h orbit overlap was decreased by 6.5 mm (10.54%), 1.8 mm (4.4%), and 1.1 mm (8.03%) in the along-track, cross-track, and radial directions, respectively; the RMS and STD of the 24 h clock offset overlap were decreased by 0.024 ns (8.6%) and 0.020 ns (13.1%), respectively; the fitting RMS of the 72 h clock offset of the quadratic polynomial was reduced by about 0.016 ns (6.5%).

**Keywords:** BeiDou-3; ECOM1; a priori solar radiation pressure model; phase center offsets; phase variations; precise orbit determination

---

## 1. Introduction

As of September 2019, 23 satellites of the BeiDou-3 satellites were launched into orbit, which are 20 medium Earth orbit (MEO) satellites, one Geostationary Earth Orbit (GEO) satellite, and two Inclined Geosynchronous Orbit (IGSO) satellites. By the end of 2020, the global constellation of BeiDou-3 will be complete, and will provide a global navigation satellite system (GNSS) positioning, navigation, and timing services, together with GPS, GLONASS, and Galileo [1,2]. Precision orbit and clock offsets in the same reference frame are a prerequisite for multi-GNSS applications. In multi-GNSS data processing, in addition to the same International Earth Rotation and Reference Systems Service (IERS) protocol, the unification of the framework is mainly reflected in the consistency of the antenna phase center correction (PCC) and the coordinates of the core station of the framework in Solution INdependent EXchange Format (SINEX). The PCC consists of the antenna phase center offsets (PCOs) and phase variations (PVs).

The International GNSS Service (IGS) [3] has released the phase center offsets (PCOs) and phase variations (PVs) for GPS, GLONASS, and Galileo satellites in igs14.atx, which is aligned to the IGS2014.

The details of the PCOs and PVs of GPS and GLONASS satellites are estimated in-orbit by GNSS tracking station data [4]. The accurate PCOs and PVs for Galileo satellites were calibrated before launch by the manufacturer and were released by the European GNSS Agency (GSA) in 2017. Before 2017, another set of PCOs of Galileo satellites was estimated by Steigenberger et al. [5]. For BeiDou-2 satellites, the PCOs and PVs of IGSO and MEO satellites were estimated in-orbit by Dilssner et al. [6,7]; a PCOs-only model was estimated by Huang et al. [8]. However, there were significant differences of about 100 cm between the estimations of the z-offset for IGSO satellites and small differences of about 20 cm for MEO satellites, and PCC models that prevented the incorporation of these results into igs08.atx [9]. The conventional Multi-GNSS Experiment (MGEX) values of the block-specific PCOs of BeiDou-2 and BeiDou-3 satellites were provided by the Test and Assessment Research Center of China Satellite Navigation Office (TARC/CSNO), which can be found in igs14_2062.atx. For the numerical values of the BeiDou-3 MGEX PCOs, the model does not cover all frequencies, and the PVs are missing; its values are set to zero. The PCC models of BeiDou-3 seem to be much rougher than Galileo's, and the accuracy of the model is unknown to the user. Thus, it is necessary to estimate the more accurate PCOs and PVs for BeiDou-3 satellites in-orbit.

The x-offset and y-offset are the horizontal components of PCOs, and the z-offset is the vertical component. The correlation coefficient between the x-offset and the solar radiation pressure (SRP) parameters can reach about 97% [8], and the error of the SRP can be absorbed by the x-offset, which can cause systematic errors regarding the elevation angle of the Sun to the plane (β-angle) in the x-offset estimation. Extending the CODE Orbit Model (ECOM1) [10] is no longer applicable to cuboid satellites, like Galileo, the Quasi-Zenith Satellite System (QZSS), and BeiDou-3 satellites [11–15], specifically, the Satellite Laser Ranging (SLR) [16,17] residuals show a systematic variation related to the β-angle. There is about 40 cm variation in the x-offset with respect to (w.r.t.) the β-angle, when using the pure ECOM1 for Galileo satellites [5]. After assistance on the a priori box-wing model developed by Montenbruck et al. [11] to ECOM1, the variation in the x-offset vanished. BeiDou-2 IGSO and MEO satellites are also fit for a cuboid satellite's body [11], although the SLR residuals of precise orbit determination (POD) do not exhibit a systematic variation w.r.t. the β-angle [18], the x-offset estimation still had a variation of 20–40 cm for IGSO satellites when ECOM1 was adopted [8]. For the BeiDou-3 MEO satellites, the SLR residuals exhibit a systematic variation of about 15 cm. Therefore, a priori box-wing models were established by Yan et al. [15]. The strong correlation between the z-offset and PVs meant PCOs were estimated by fixing the PVs; then, the "raw" PVs were determined when fixing the PCOs, and subsequently, the correction of z-offset and PVs were derived by a separate least square adjustment [19–21].

This study aims to estimate high accurate PCOs and PVs for BeiDou-3 MEO satellites in orbit using GNSS ground tracking data. First, basic models of PCOs and PVs estimation, an SRP model, and a priori box-wing model are introduced. Next, based on the GNSS data, the horizontal PCOs are estimated, and the z-offset and PVs parameters are estimated separately. Orbit overlaps and clock offset overlaps precision is selected to validate the accuracy of the PCOs and PVs of the BeiDou-3 MEO. Meanwhile, the performance assessment is compared between the IGS PCOs. Finally, a summary of this work is given.

## 2. Basic Models

PCOs and PVs parameters must be obtained separately. Either PCOs or PVs were estimated by POD together with satellite orbit parameters, clock offsets, tropospheric zenith delay (ZTD), station coordinates, ambiguity, inter-system biases, and earth rotation parameters (ERPs). The function models of PCOs and PVs were given in this part. As part of the satellite orbital parameters, the SRP model of the GNSS POD was also introduced in this work. Considering the correlation between the SRP and x-offset, an a priori box-wing model of BeiDou-3 MEO satellite is shown in this section.

*2.1. PCO Parameters Model*

The satellite antenna PCO is a vector from the mass center to phase center, which consists of x-offset, y-offset, and z-offset in a satellite body-fixed coordinate system [22]. The correction of PCO parameters on the observed distance is shown in Equation (1):

$$\Delta\rho(\alpha, \eta) = dx \cdot \sin\alpha\sin\eta + dy \cdot \cos\alpha\sin\eta + dz \cdot \cos\eta \tag{1}$$

where $\alpha$ is the azimuth and $\eta$ is the nadir angle seen from the satellite to the station, and the azimuth $\alpha$ is started from the y-axis toward the x-axis of the satellite body-fixed system when viewing from the station. $dx$, $dy$, and $dz$ are x-offset, y-offset, and z-offset, respectively.

*2.2. PV Parameters Model*

Due to the high correlation between the satellite antenna PVs and z-offset, raw PVs were estimated that correspond to the following sum [19,21]:

$$PVraw(\eta) = PV(\eta) + \Delta z \cdot (1 - \cos\eta) \tag{2}$$

where $\Delta z$ was the correction of the a priori value of the z-offset, to prevent the normal equation from being singular and considering the correlation with the satellite clock offsets, an a priori constraint is added:

$$\sum_{i=0}^{n} PVraw(\eta_i) = 0 \tag{3}$$

The $PVraw(\eta_i)$ are estimated satellite-by-satellite as piece-wise constants. After $PVraw(\eta_i)$ parameters are obtained, a separate adjustment model, shown as Equation (2), is established to derive $PV(\eta_i)$ and $\Delta z$ parameters. The criterion of least square adjustment is as follows:

$$\sum_{i=0}^{n} \left[ PVraw(\eta_i) - a - \Delta z(1 - \cos\eta_i) \right]^2 = \min \tag{4}$$

where $a$ is a constant parameter, and $n$ is the maximum integer nadir-angle of $PV(\eta_i)$. The residuals of the least square adjustment are PVs, and the PVs and z-offset datum is $\sum_{i=0}^{n} PV(\eta_i) = 0$. This model mainly used for GPS and GLONASS to estimate satellites phase center correction [4,19].

*2.3. Solar Radiation Pressure Model*

The empirical Center for Orbit Determination in Europe (CODE) orbit model is expressed in a Sun-oriented reference frame, where $D$ points to the Sun, $Y$ goes along with the solar panel axis, and $B$ completes a right-handed system [10]. The ECOM1 model can be expressed as:

$$\begin{aligned} D &= D_0 \\ Y &= Y_0 \\ B &= B_0 + B_c \cdot \cos u + B_s \cdot \sin u \end{aligned} \tag{5}$$

where $D_0$, $Y_0$, $B_0$ are three constant parameters, $B_c$ and $B_s$ are cosine and sine terms in $B$ direction, and $u$ is the argument of latitude. ECOM1 was widely adopted for the SRP of GPS and GLONASS for most of the Multi-GNSS Experiment (MGEX) analysis centers [7,23].

Although the ECOM2 model [24] was better used than the ECOM1 for precise orbit determination of the cuboid satellites [15,25], it was not applied for estimating PCOs, because the increased parameters of second- and fourth-order harmonic terms for the D-component caused a higher correlation between PCOs and SRP parameters [5].

### 2.4. A Priori Box-Wing Model for BeiDou-3 MEO

During the nominal yaw-steering period [22], the box-wing model is reformulated as Equation (6) [11]:

$$
\begin{aligned}
a_D = \quad & -a_1 \cdot (|\cos \varepsilon| + \sin \varepsilon + \tfrac{2}{3}) \\
& -a_2 \cdot (|\cos \varepsilon| - \sin \varepsilon - \tfrac{4}{3}\sin^2 \varepsilon + \tfrac{2}{3}) \\
& -a_3 \cdot (\cos \varepsilon + \tfrac{2}{3}|\cos \varepsilon| \cos \varepsilon) \\
& -a_4 \cdot 2(|\cos \varepsilon| \cos^2 \varepsilon + \sin^3 \varepsilon) \\
& -a_5 \cdot 2(|\cos \varepsilon| \cos^2 \varepsilon - \sin^3 \varepsilon) \\
& -a_6 \cdot 2(\cos^3 \varepsilon) \\
& -a_7
\end{aligned}
\tag{6}
$$

$$
\begin{aligned}
a_B = \quad & -a_2 \cdot \tfrac{4}{3}(\cos \varepsilon \sin \varepsilon) \\
& -a_3 \cdot \tfrac{2}{3}(|\cos \varepsilon| \sin \varepsilon) \\
& -a_4 \cdot 2((|\cos \varepsilon| - \sin \varepsilon) \cos \varepsilon \sin \varepsilon) \\
& -a_5 \cdot 2((|\cos \varepsilon| + \sin \varepsilon) \cos \varepsilon \sin \varepsilon) \\
& -a_6 \cdot 2(\cos^2 \varepsilon \sin \varepsilon)
\end{aligned}
\tag{7}
$$

where $a_D$ and $a_B$ are the accelerations in D and B directions, respectively; $\varepsilon$ is the sun–satellite–earth angle; $a_1, a_2, a_3, a_4, a_5, a_6, a_7$ are the model coefficients.

The a priori box-wing model can provide a priori value to the pure ECOM1 model. It can impose reasonable constraints on the ECOM1 parameters when estimating PCOs and obtain more stable x-offset parameters. There are two satellite manufacturers for the BeiDou-3 satellites: the China Academy of Space Technology (CAST) and the Shanghai Engineering Center for Microsatellites (SECM) [26,27]. The optical properties and shape sizes of the two types of satellites are different, so the coefficients of the a priori box-wing model are also different. The a priori box-wing model of the two types of BeiDou-3 MEO satellites was estimated by Yan et al. [15], and the coefficients are listed in Table 1.

**Table 1.** Coefficients of a priori box-wing model for the BeiDou-3 medium Earth orbit (MEO) satellites, the unit is nm/s$^2$.

| Types | $a_1$ | $a_2$ | $a_3$ | $a_4$ | $a_5$ | $a_6$ | $a_7$ |
|-------|-------|-------|-------|-------|-------|-------|-------|
| CAST  | 5.99  | -0.32 | -1.18 | 11.10 | -0.53 | 0.21  | 110.62 |
| SECM  | 3.02  | 1.05  | 0.84  | 5.61  | 1.95  | 0.05  | 59.01  |

## 3. Strategies

The distribution map of the stations of precise orbit determination for the BeiDou-3 satellites is shown in Figure 1. Among them, 56 MGEX/IGS stations and 17 international GNSS Monitoring and Assessment System (iGMAS) stations provide BeiDou-3 and GPS observations.

The general data processing strategies for estimating PCOs, PVs, and precise orbit determination of the BeiDou-3 satellites are listed in Table 2, mainly including observation models, error models, and parameter estimation models.

For special data processing, the differences will be described before the analysis of results. The conventional MGEX values of the block-specific PCOs of the BeiDou-3 satellites are listed in Table 3.

There are four frequencies, named C01, C02, C06, and, C07, in the conventional MGEX PCOs model. Duplicate entries for C01 and C02 support the use of RINEX 3.01 and 3.02 observation codes of B1I, and the code of the subsequent RINEX versions, respectively. The conventional MGEX PCOs are both the initial value when they were estimated, and the reference model when the estimated PCOs were validated.

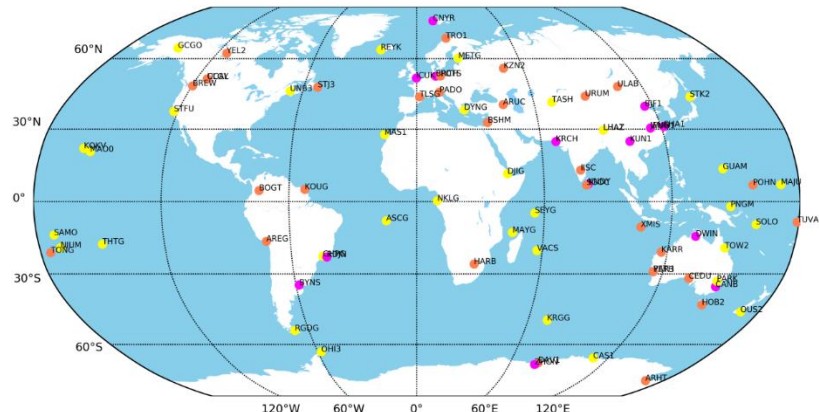

**Figure 1.** Distributions of precise orbit determination (POD): magenta points are international GNSS Monitoring and Assessment System (iGMAS) stations, which can track all BeiDou-2/3 satellites; orange and yellow points are MGEX stations, of which orange points can provide BeiDou-3 data.

**Table 2.** Strategies for data processing.

| Items | Descriptions |
|---|---|
| Stations | About 17 iGMAS stations and 56 IGS/MGEX stations; |
| Time period | From 214, 2018 to 140, 2019; |
| Observation | Zero-difference phase and code observations;<br>Elevation-dependent weight; elevation cutoff angle is 15°; |
| Data arc | 72 h orbital arcs; |
| Attitude model | Yaw-steering mode; |
| Solar radiation pressure (SRP) | ECOM1+a priori box-wing model (Table 1); |
| Inter-system biases (ISBs) | A constant parameter for each station per orbital arc, and zero-mean constraints were added for all ISBs; |
| Ionosphere delay | Ionosphere-free linear combination<br>GPS: L1/L2<br>BeiDou-3: B1I/B3I; |
| Troposphere delay | ZTD parameters with an interval of 2 h;<br>SAAS + GMF [28–30];<br>horizontal gradient parameters with an interval of 24 h; |
| Station coordinates | Fixed to the IGSYYPWWWW.snx, where YY is the last two digits of the year, and WWWW is the corresponding GPS week; |
| Receiver antenna | Fixed to igs14_WWWW.atx; |
| Satellite antenna | GPS and BeiDou satellites are from igs14_2062.atx; |
| Ambiguity | Fixed by adding double-difference constraint [31,32]; |
| Eclipsing period | Removed. |

**Table 3.** Multi-GNSS Experiment (MGEX) phase center offsets (PCOs) of BeiDou-3 satellites refer to igs14_2062.atx, unit is mm.

| Frequency | CAST | | | SECM | | |
|---|---|---|---|---|---|---|
| | x-offset | y-offset | z-offset | x-offset | y-offset | z-offset |
| C01 | −200 | 0 | 1460 | 40 | −10 | 1100 |
| C02 | −200 | 0 | 1460 | 40 | −10 | 1100 |
| C06 | −200 | 0 | 1180 | 40 | −10 | 1090 |
| C07 | −200 | 0 | 1070 | 40 | −10 | 1090 |
| Iono-free combination of B1I and B3I | −200 | 0 | 2004 | 40 | −10 | 1119 |



## 4. PCOs and PVs Estimations

Considering that the factors affecting the three components are different, we divided the PCOs into horizontal and vertical PCOs. The horizontal PCOs are composed of the x-offset and y-offset, which are affected by the SRP model. Here, the impact of different SRP models on the x-offset are analyzed.

### 4.1. Horizontal PCO Estimations

When PCOs were estimated by the pure ECOM1 (Figure 2a), the x-offset time series of the C21 satellite produced a systematic variation of about 40 cm with the β-angle, which is similar to the results of the Galileo satellites [5]. Therefore, the a priori box-wing model (Table 1) was used to assist the ECOM1 model (Figure 2b). Although more stable x-offset estimations were obtained, there were still large scatters when the absolute value of the β-angle was large. Further, when a constraint of 1 nm/s$^2$ was added to the $D_0$ parameter (Figure 2c), the x-offset series appeared to be a stationary sequence that was almost irrelevant to the β-angle.

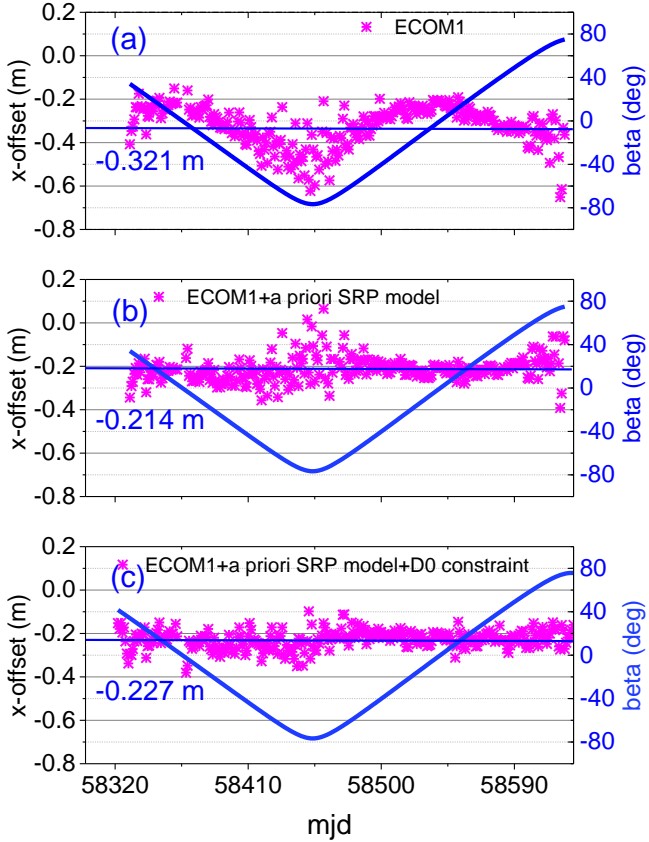

**Figure 2.** The x-offset estimations of the C21 using different solar radiation pressure (SRP) models. Figure (**a**) is the pure ECOM1 model, Figure (**b**) is the ECOM1+a priori SRP model, and Figure (**c**) is the ECOM1+ a priori SRP model with a constraint of 1 nm/s$^2$ for the $D_0$ parameter. The blue line is the β-angle.

Comparing the average x-offset of the three SRP models (Figure 2a–c), it was found that there was a significant difference of about 11 cm between the pure ECOM1 and the a priori model-assisted ECOM1 solutions. For the results using the a priori model, the difference between the constraint and non-constraint was 1.3 cm. For the other CAST satellites, similar results were found. However, no significant systematic variations in the x-offset were found for SECM satellites, such as C25, C26, C27, C28, C29, and C30, which had smaller ranges of β-angle of ±60° and ±40°. For the three SRP models, there was no significant effect on the y-offset time series.

To verify the impact of the $D_0$ constraint on the precision of the orbit overlap when estimating the PCOs, three schemes of experiments were designed: (a) the precision of the overlap without estimating PCOs (Figure 3a); (b) estimating PCOs but not adding a constraint to the $D_0$ parameter (Figure 3b); (c) estimating PCOs and adding a constraint of 1 nm/s$^2$ to the $D_0$ parameter (Figure 3c). In Figure 2b, corresponding to the larger scatters of x-offset, $|\beta| > 60°$, the results were the gray-shaded area in Figure 3, and the RMS of the orbit overlap in along-track (A), cross-track (C), and radial (R) directions were computed and shown in Table 4. Compared with the results of estimating PCOs and non-constraint of $D_0$ parameter in scheme (b), the RMS of scheme (c) was reduced by 0.65 (6.8%), 6.36 (59.2%), and 0.4 (16.3%) cm, in A-, C-, and R-components, respectively. Therefore, it can be inferred that the correlation between the x-offset and the C-component of the orbit was higher than the A- and R-components for $|\beta| > 60°$. After adding the constraint to the $D_0$ parameter, the RMS of orbit overlap was close to the reference result (scheme a), and the differences between scheme (c) and scheme (a) were 0.79, −1.72, and 0.25 cm, for A-, C-, and R-components, respectively.

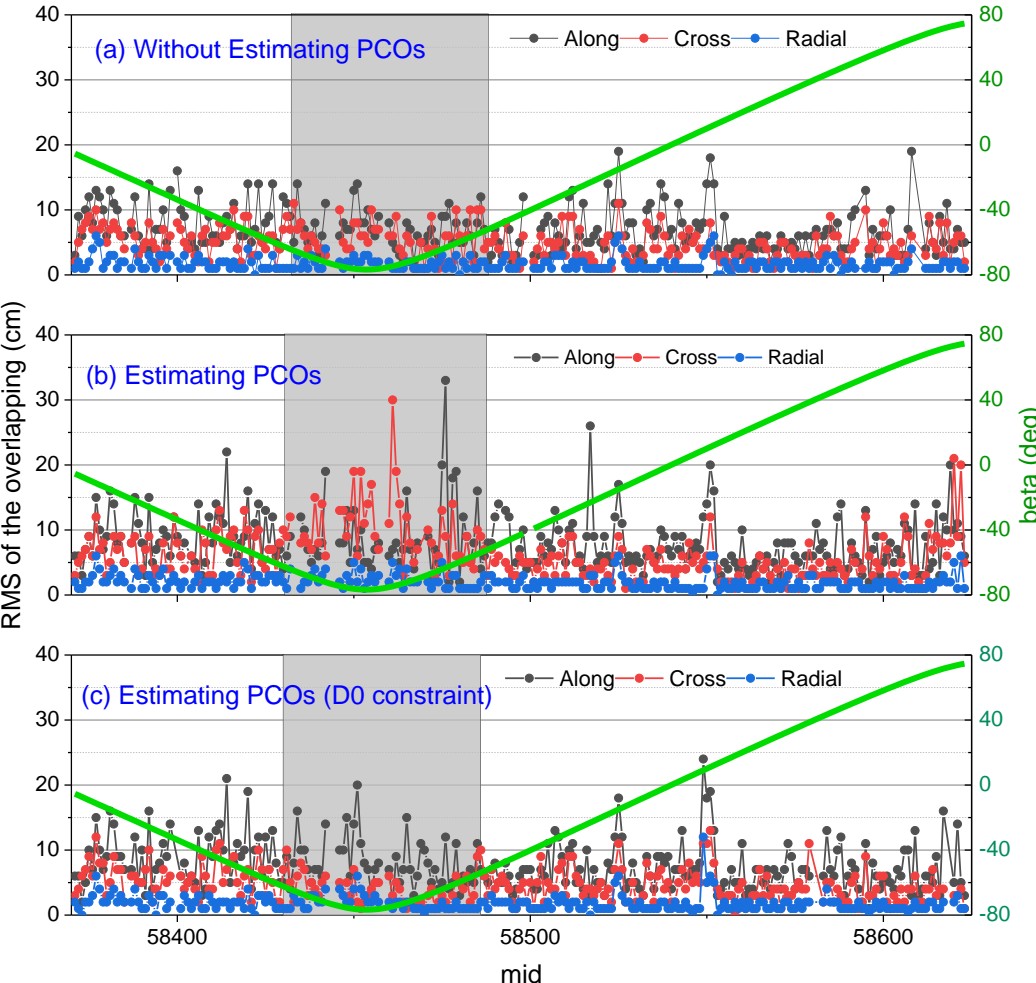

**Figure 3.** The RMS of the overlap of the C21 satellite. Figure (**a**) is the result without estimating PCOs, Figure (**b**) is the result estimating PCOs without adding a constraint for the $D_0$ parameter, and Figure (**c**) is the result estimating PCOs with a constraint of 1 nm/s$^2$ for the $D_0$ parameter. The blue line is the β-angle. The black, red, and black points are along-, cross-, and radial-components, respectively. The gray-shaded area was the result for about $|\beta| > 60°$, where the RMS was significantly larger than others in Figure (**b**).

**Table 4.** The average RMS of the orbit overlap, the unit is cm.

| Scheme | A | C | R |
|---|---|---|---|
| (a) Without estimating PCOs | 8.08 | 6.10 | 1.81 |
| (b) Estimating PCOs, without $D_0$ constraint | 9.52 | 10.74 | 2.46 |
| (c) Estimating PCOs, with $D_0$ constraint of 1 nm/s$^2$ | 8.87 | 4.38 | 2.06 |

After obtaining the daily horizontal PCOs time series, the average of each satellite was calculated as the estimation of this satellite. The standard deviation (STD) was also used as an indicator to measure the repeatability of the daily PCOs series. The specific results are shown in Table 5. From the STD values, a centimeter precision level of horizontal PCOs was obtained. Considering that the PCOs of the same type of satellite are close, such as C19–C24 for CAST satellites, and C25–C30 for SECM, to improve the reliability of the PCOs, the average of the same type of satellite was taken as the final type-specific satellite PCOs. The STD of inner-type PCOs was also calculated. Compared with other satellites, C32, C33, and C35 were launched into orbit relatively late and had a short period of valid observations, resulting in poor accuracy of PCOs, the three satellites were excluded when calculating the final type-specific PCOs. The final PCOs and STDs are shown in Table 6.

**Table 5.** Average and standard deviation (STD) of the horizontal PCOs for each satellite, the unit is mm.

| PRN | x-offset | | y-offset | |
|---|---|---|---|---|
| | Average | STD | Average | STD |
| C19 | −228.28 | 33 | −6.09 | 59 |
| C20 | −229.22 | 40 | −15.77 | 60 |
| C21 | −211.47 | 44 | −8.56 | 61 |
| C22 | −226.40 | 38 | −13.86 | 51 |
| C23 | −230.12 | 39 | 1.74 | 52 |
| C24 | −218.68 | 43 | −11.19 | 51 |
| C25 | 37.69 | 48 | −12.54 | 62 |
| C26 | 29.26 | 55 | −4.01 | 64 |
| C27 | 18.08 | 49 | −7.54 | 33 |
| C28 | 18.99 | 29 | −4.64 | 19 |
| C29 | 19.46 | 36 | −11.63 | 38 |
| C30 | 21.29 | 35 | −6.25 | 24 |
| C32 | −165.81 | 63 | −8.00 | 52 |
| C33 | −164.22 | 63 | −8.99 | 40 |
| C34 | 44.66 | 30 | −6.63 | 21 |
| C35 | 126.23 | 57 | −9.28 | 40 |

**Table 6.** Final estimations and STDs of the horizontal PCOs for the BeiDou-3 MEO, the unit is mm.

| Type | x-offset | | y-offset | |
|---|---|---|---|---|
| | Average | STD | Average | STD |
| CAST | −224.48 | 6 | −9.34 | 7 |
| SECM | 24.00 | 10 | −7.81 | 4 |

### 4.2. Vertical PCO and PV Estimations

The main factors affecting the z-offset are the PVs and satellite clock offset. Thus, the PVs and z-offset were estimated in two steps. Firstly, the z-offset was estimated while the PVs were fixed as zero for all nadir-angles, then, the z-offset was fixed as the estimation, and the PVraws were estimated as a linear piece-wise constant model. Subsequently, PVs were determined by removing the cosine variation and the constant according to Equation (4), which yielded final PVs and correction to the z-offset.

The z-offset estimations and the STD values of BeiDou-3 MEO satellites in the first step are listed in Table 7 when the PVs were fixed to the value of zero. The z-offset estimations of C21 and C30 were selected as the representatives of CAST and SECM satellites and are shown in Figure 4. For the CAST satellites, the largest difference in the z-offset was 156 mm and the average value of STD was 124 mm, which indicated a good consistency between z-offsets of satellites. For the SECM satellites, except for the C25, C29, and C35 satellites, the maximum difference of z-offset was 139 mm and the average value of the STD was 121mm, which indicated a good consistency of z-offset between the same type of satellite. Therefore, the average values of 2368.38 and 1596.38 mm were fixed as the z-offset estimations to solve PVs for CAST and SECM satellites, respectively.

**Table 7.** Average and STD of the z-offset for each satellite, the unit is mm.

| CAST | C19 | C20 | C21 | C22 | C23 | C24 | C32 | C33 |
|------|-----|-----|-----|-----|-----|-----|-----|-----|
| z-offset | 2300.6 | 2339.1 | 2310.4 | 2333.1 | 2370.9 | 2426.5 | 2406.9 | 2459.5 |
| STD | 126.9 | 128.9 | 109.8 | 121.5 | 127.2 | 136.0 | 117.8 | 119.0 |
| **SECM** | **C25** | **C26** | **C27** | **C28** | **C29** | **C30** | **C34** | **C35** |
| z-offset | 1429.1 | 1542.0 | 1627.3 | 1680.8 | 1739.7 | 1564.8 | 1567.0 | 1445.2 |
| STD | 147.1 | 139.3 | 173.5 | 97.9 | 121.7 | 105.7 | 90.9 | 193.6 |

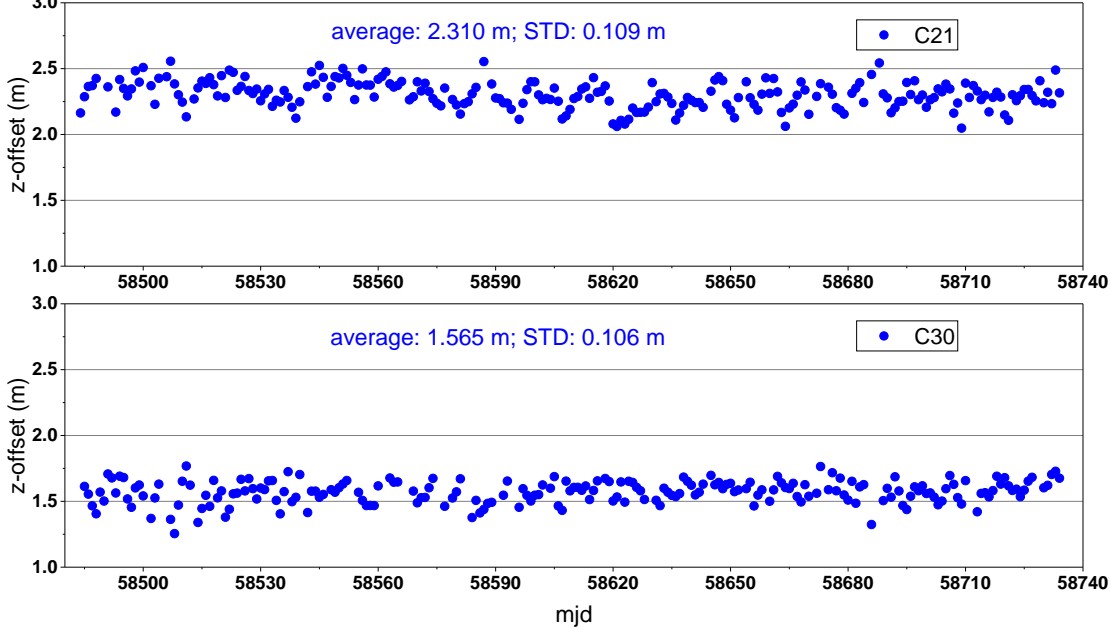

**Figure 4.** Estimations of the z-offset of the C21 (**top**) and C30 (**bottom**) satellites.

For each satellite, PVraws were obtained from the average of the daily PVraws. The day repeatability was 1–2 mm at the 0° nadir-angle, 1 mm at 1° and 13°, and less than 1 mm for the other nadir-angles. The PVraws of each satellite are shown in Figure 5. For the average PVraws of all satellites, the STD was 1.6 mm at the 0° nadir-angle, 1 mm at 1° and 13°, and less than 1 mm for the others. The reasons were mainly that the observations at 0 and 1° nadir angles were few, and the weight of the observations at the 13° was reduced.

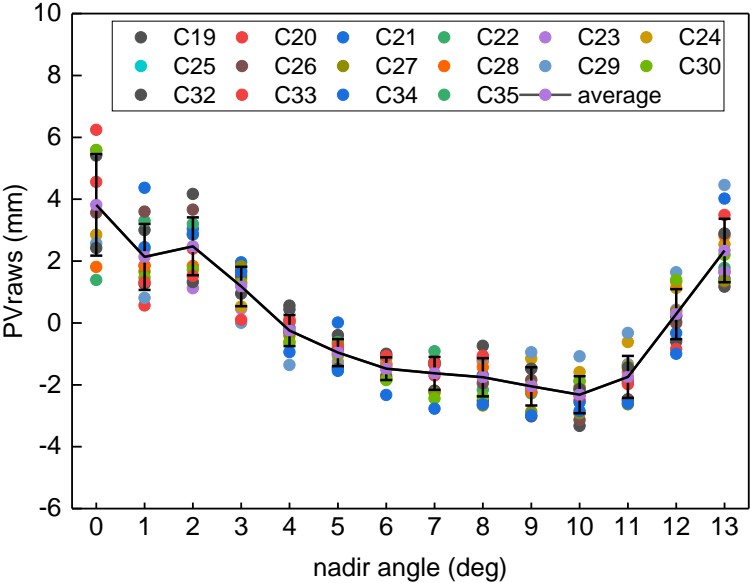

**Figure 5.** The PVraws and repeatability of the BeiDou-3 MEO satellites.

Based on Equation (4), a separate least-squares adjustment was carried out to derive the unmodeled z-offset and PVs. After the adjustment, the PVs are shown in Figure 6a. Then, the average corrections of z-offsets are shown in Figure 6b. The final PVs and the z-offsets of the BeiDou-3 MEO satellites are listed in Tables 8 and 9. Considering that the z-offsets were different from other satellites, the estimations of C25 and C29 were given separately, and the type-specific z-offsets and STDs were given. For the CAST and SECM, the z-offset estimations had a good internal consistency with STDs of 35 and 51 mm, respectively.

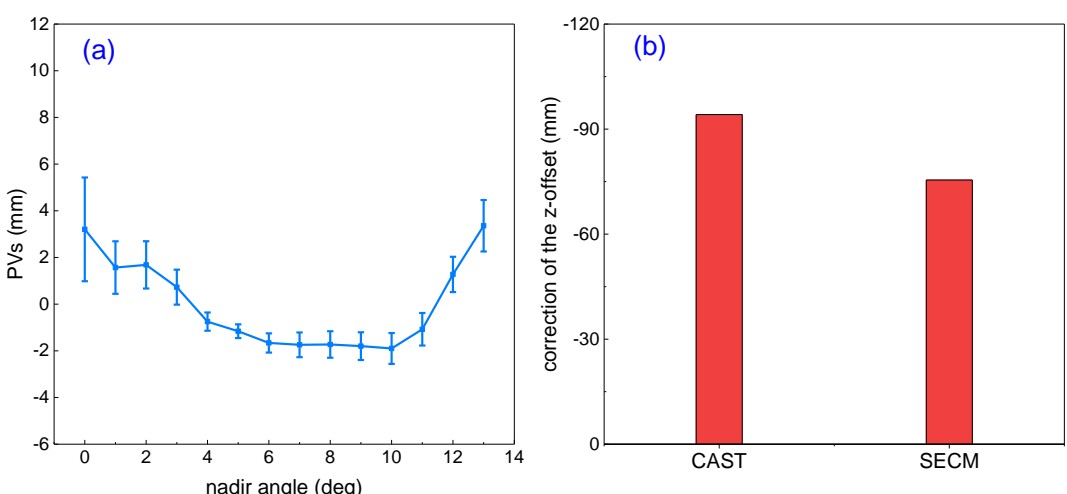

**Figure 6.** The PVs (**a**) and correction of the z-offset (**b**) estimations.

**Table 8.** Final PVs and STDs of the BeiDou-3 MEO satellite, the unit is mm.

| nadir angle/° | 0 | 1 | 2 | 3 | 4 | 5 | 6 |
|---|---|---|---|---|---|---|---|
| PVs | 3.20 | 1.57 | 1.68 | 0.73 | −0.75 | −1.16 | −1.66 |
| STD | 2.22 | 1.13 | 1.01 | 0.75 | 0.39 | 0.29 | 0.41 |
| **nadir angle/°** | **7** | **8** | **9** | **10** | **11** | **12** | **13** |
| PVs | −1.74 | −1.73 | −1.80 | −1.90 | −1.08 | 1.27 | 3.36 |
| STD | 0.53 | 0.57 | 0.59 | 0.66 | 0.70 | 0.76 | 1.10 |

**Table 9.** Final estimations and STDs of z-offset for the BeiDou-3 MEO, the unit is mm.

| Type | z-offset | STD |
|---|---|---|
| CAST | 2274.22 | 35 |
| SECM | 1520.94 | 51 |
| C25 | 2341.41 | - |
| C29 | 1695.62 | - |

## 5. Validations

To verify the impact of different satellite antenna phase center correction models on the precise orbit determination and clock offset estimation, three inspectors of overlap orbit, overlap clock offset, and clock offset fitting were selected. Three schemes were designed for comparison:

Scheme 1: Precise orbit determination by the conventional MGEX PCOs model (Table 3), named S1;

Scheme 2: Precise orbit determination by the estimated PCOs-only model (Table 9), named S2;

Scheme 3: Precise orbit determination by the estimated PCOs and PVs models (Tables 8 and 9), named S3;

The selected time was from the day of the year (doy) of 001 to 180 in 2019. Precise orbit determination was implemented according to schemes 1, 2, and 3, respectively. Compared to Table 2, the difference in the strategy was the PCC of the BeiDou-3 MEO satellites.

### 5.1. Orbit Precision

Since the 3-day arcs were adopted for the precision orbit determination, the difference in the overlap between the adjacent two solutions can be used to measure the internal coincidence of the orbit. Specifically, the orbit of the middle day of the 3-day solution is taken as a reference to measure the precision of the third day of the previous 3-day solution [33,34], and the overlap RMS values are given in along-track (A), cross-track (C), and radial (R) directions, are shown in Table 10.

**Table 10.** Precision of 24 h orbit overlap, unit is cm.

| PRN | S1 | | | S2 | | | S3 | | |
|---|---|---|---|---|---|---|---|---|---|
| | A | C | R | A | C | R | A | C | R |
| C19 | 5.76 | 3.71 | 1.21 | 5.02 | 3.43 | 1.09 | 4.99 | 3.44 | 1.03 |
| C20 | 6.06 | 3.64 | 1.36 | 5.35 | 3.48 | 1.18 | 5.04 | 3.51 | 1.14 |
| C21 | 5.50 | 3.70 | 1.35 | 4.98 | 3.36 | 1.04 | 5.04 | 3.53 | 1.28 |
| C22 | 5.60 | 3.73 | 1.20 | 5.09 | 3.50 | 1.14 | 4.86 | 3.40 | 1.15 |
| C23 | 6.62 | 4.43 | 1.64 | 6.26 | 4.15 | 1.55 | 6.01 | 3.98 | 1.51 |
| C24 | 6.83 | 4.51 | 1.64 | 6.49 | 4.54 | 1.59 | 6.34 | 4.44 | 1.47 |
| C25 | 7.23 | 5.25 | 1.67 | 6.19 | 4.93 | 1.54 | 6.42 | 5.04 | 1.61 |
| C26 | 6.74 | 4.52 | 1.47 | 6.56 | 4.50 | 1.44 | 7.08 | 4.60 | 1.53 |
| C27 | 6.98 | 4.31 | 1.37 | 6.55 | 4.10 | 1.22 | 6.11 | 4.05 | 1.20 |
| C28 | 6.63 | 3.89 | 1.18 | 5.26 | 3.58 | 0.88 | 5.12 | 3.55 | 0.85 |
| C29 | 7.14 | 4.30 | 1.31 | 6.27 | 4.29 | 1.11 | 6.24 | 4.32 | 1.10 |
| C30 | 6.47 | 4.39 | 1.21 | 5.63 | 4.27 | 1.11 | 5.87 | 4.23 | 1.11 |
| C32 | 6.88 | 4.43 | 1.43 | 5.98 | 4.22 | 1.36 | 6.76 | 4.31 | 1.50 |
| C33 | 6.79 | 4.59 | 1.47 | 6.53 | 4.35 | 1.39 | 6.71 | 4.50 | 1.53 |
| C34 | 6.81 | 4.04 | 1.09 | 6.05 | 3.67 | 1.00 | 5.71 | 3.77 | 1.01 |
| C35 | 10.16 | 6.34 | 2.76 | 10.20 | 6.55 | 2.68 | 9.52 | 6.16 | 2.61 |
| Average | 6.76 | 4.36 | 1.46 | 6.15 | 4.18 | 1.33 | 6.11 | 4.18 | 1.35 |

It can be seen from Table 10 that the precision of the orbit overlap of the C35 satellite was worse than that of other satellites, in which the RMS of A-direction reached 10 cm, and the radial direction also reached 2.7 cm. The observations of the C35 were less than the others, resulting in poor accuracy of precise orbit determination. The averages of RMS in the three schemes were computed. Compared with

the MGEX PCOs (S1), the precision of the overlap with the estimated PCOs-only (S2) was increased by 6.1 mm (9.08%), 1.8 mm (4.10%), and 1.3 mm (8.64%) in the A-, C-, and R-directions, respectively; the precision of the overlap using the estimated PCOs and PVs (S3) increased by 6.5 mm (10.54%), 1.8 mm (4.4%), and 1.1 mm (8.03%) in the A-, C-, and R-directions, respectively. The improvement in the A-direction was more significant than the C- and R- directions when the estimated PCOs were used.

To quantitatively analyze the number of effective observations, and the contribution of the MGEX and iGMAS networks for the BeiDou-3 satellites, 36 and 14 stations were selected from MGEX and iGMAS, respectively. The time was randomly selected as day 150 of the year. The number of effective observations after POD is analyzed in Figure 7. It can be seen that C19, C20, C21, C22, and C28 had the most observations, with about 8300 from MGEX, and about 2900 from iGMAS. C23–C27 and C29–C34 had about 4500 observations from MGEX and 2700 from iGMAS. C35 had a smaller number of observations, with about 690 from MGEX. An expected ratio of the number of observations of MGEX and iGMAS was about 2.57 (36/14). A measured ratio of the number of observations of MGEX and iGMAS was computed by the actual number after POD. C23–C27 and C29–C34 had a lower magnitude (measured ratio < expected ratio) for about 1.57 of the number of observations. In particular, C35 had the lowest measured ratio of 0.23. The loss of the pseudorange and phase measurements of the B3I frequency mainly caused a lower measured ratio than expected. C19, C20, C21, C22, and C28 had a normal level of number of observations from the MGEX, with a measured ratio of 2.75. The main reason for this is that the iGMAS receiver has undergone software and hardware updates and has a better ability to receive BeiDou-3 satellite observations. However, the receivers of most of the MGEX stations have not been updated, and some of the receivers that have been replaced or updated have maintained good reception capabilities during the data processing period in this study. As of November 2019, after most of the MGEX receivers have been upgraded, data reception has returned to normal.

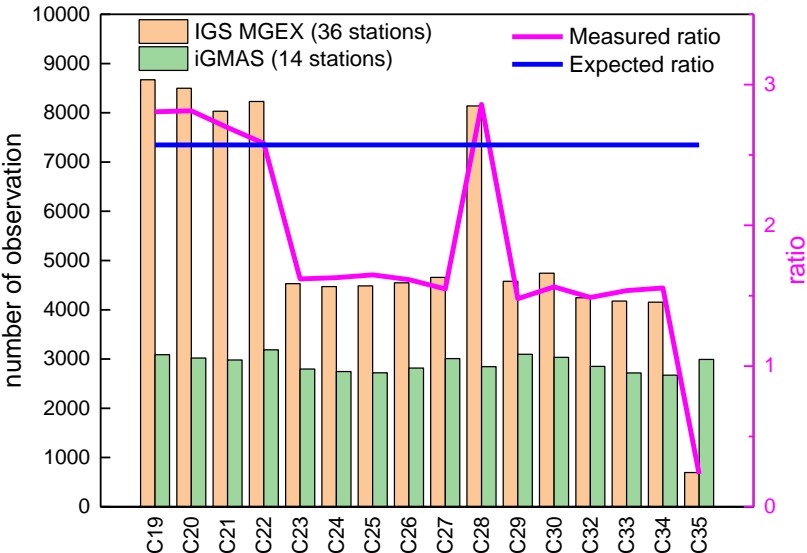

**Figure 7.** The number of observations of the BeiDou-3 satellites. Orange and green bars are the number of observations provided by the MGEX and iGMAS networks, respectively; magenta and blue lines are the measured and expected ratios of MGEX to iGMAS, respectively.

*5.2. Clock Offset Precision*

The precision of the 24 h clock offset overlap was taken to evaluate the accuracy of different PCC models (S1, S2, and S3). The STD and RMS were obtained by a quadratic difference between the satellite clock offset series and reference clock offset series [35], where the reference clock of BREW was chosen. The specific clock offset precision is shown in Table 11. The RMS and STD of the C35 satellite were 0.34 ns and 0.25 ns, which showed a poorer accuracy than other satellites. One possible

reason was the lack of observations (seen from Figure 7), another is a poorer accuracy of PCOs of the C35 compared to other satellites of SECM. After getting sufficient observations, further analysis of the specific reasons for this is required. Compared to the MGEX PCOs result (S1), the average RMS and STD of the estimated PCOs-only (S2) were decreased by 0.016 ns (5.8%) and 0.017 ns (11.3%), respectively. For the estimated PCOs and PVs result (S3), the RMS and STD of the clock offset were decreased by 0.024 ns (8.6%) and 0.020 ns (13.1%), respectively. The RMS of quadratic polynomial fitting of 72 h clock offset overlap is shown in Table 12.

**Table 11.** Precision of 24 h clock offset overlap, unit is ns.

| PRN | S1 | | S2 | | S3 | | S1-S2 | | S1-S3 | |
| --- | --- | --- | --- | --- | --- | --- | --- | --- | --- | --- |
| | RMS | STD | RMS | STD | RMS | STD | RMS | STD | RMS | STD |
| C19 | 0.251 | 0.120 | 0.228 | 0.113 | 0.228 | 0.107 | 0.023 | 0.007 | 0.022 | 0.012 |
| C20 | 0.255 | 0.114 | 0.232 | 0.115 | 0.216 | 0.105 | 0.024 | −0.001 | 0.040 | 0.009 |
| C21 | 0.240 | 0.121 | 0.237 | 0.114 | 0.223 | 0.109 | 0.003 | 0.007 | 0.017 | 0.012 |
| C22 | 0.249 | 0.126 | 0.241 | 0.125 | 0.236 | 0.113 | 0.008 | 0.001 | 0.013 | 0.013 |
| C23 | 0.290 | 0.166 | 0.271 | 0.148 | 0.270 | 0.149 | 0.019 | 0.017 | 0.020 | 0.017 |
| C24 | 0.275 | 0.149 | 0.264 | 0.134 | 0.264 | 0.140 | 0.011 | 0.015 | 0.011 | 0.009 |
| C25 | 0.294 | 0.175 | 0.279 | 0.144 | 0.268 | 0.151 | 0.015 | 0.031 | 0.026 | 0.024 |
| C26 | 0.301 | 0.174 | 0.281 | 0.136 | 0.266 | 0.141 | 0.020 | 0.038 | 0.035 | 0.033 |
| C27 | 0.299 | 0.150 | 0.285 | 0.119 | 0.253 | 0.118 | 0.014 | 0.031 | 0.046 | 0.032 |
| C28 | 0.267 | 0.120 | 0.240 | 0.100 | 0.234 | 0.094 | 0.026 | 0.020 | 0.033 | 0.027 |
| C29 | 0.284 | 0.146 | 0.255 | 0.129 | 0.258 | 0.121 | 0.029 | 0.017 | 0.026 | 0.025 |
| C30 | 0.285 | 0.134 | 0.265 | 0.116 | 0.261 | 0.113 | 0.020 | 0.019 | 0.024 | 0.022 |
| C32 | 0.293 | 0.156 | 0.286 | 0.146 | 0.278 | 0.144 | 0.007 | 0.010 | 0.015 | 0.012 |
| C33 | 0.290 | 0.151 | 0.258 | 0.143 | 0.258 | 0.133 | 0.033 | 0.008 | 0.032 | 0.018 |
| C34 | 0.272 | 0.150 | 0.273 | 0.126 | 0.258 | 0.120 | −0.001 | 0.024 | 0.014 | 0.030 |
| C35 | 0.340 | 0.248 | 0.329 | 0.220 | 0.329 | 0.228 | 0.011 | 0.028 | 0.011 | 0.020 |
| Average | 0.280 | 0.150 | 0.264 | 0.133 | 0.256 | 0.130 | 0.016 | 0.017 | 0.024 | 0.020 |

**Table 12.** Precision of quadratic polynomial fitting of 72 h clock offset overlap, unit is ns.

| PRN | S1 | S2 | S3 | S1-S2 | S1-S3 |
| --- | --- | --- | --- | --- | --- |
| C19 | 0.254 | 0.244 | 0.238 | 0.010 | 0.015 |
| C20 | 0.242 | 0.237 | 0.238 | 0.004 | 0.004 |
| C21 | 0.302 | 0.299 | 0.297 | 0.003 | 0.006 |
| C22 | 0.303 | 0.298 | 0.292 | 0.005 | 0.010 |
| C23 | 0.252 | 0.237 | 0.233 | 0.016 | 0.020 |
| C24 | 0.243 | 0.245 | 0.232 | -0.002 | 0.011 |
| C25 | 0.258 | 0.250 | 0.236 | 0.008 | 0.021 |
| C26 | 0.233 | 0.214 | 0.210 | 0.019 | 0.022 |
| C27 | 0.213 | 0.203 | 0.187 | 0.010 | 0.026 |
| C28 | 0.153 | 0.154 | 0.141 | 0.000 | 0.013 |
| C29 | 0.172 | 0.168 | 0.165 | 0.004 | 0.007 |
| C30 | 0.171 | 0.165 | 0.162 | 0.006 | 0.009 |
| C32 | 0.196 | 0.192 | 0.190 | 0.004 | 0.006 |
| C33 | 0.238 | 0.231 | 0.233 | 0.006 | 0.005 |
| C34 | 0.269 | 0.221 | 0.218 | 0.048 | 0.051 |
| C35 | 0.394 | 0.383 | 0.368 | 0.011 | 0.026 |
| Average | 0.243 | 0.234 | 0.227 | 0.009 | 0.016 |

From Table 12, the fitted RMS values of the CAST satellites (C19–C24, C32, C33) were 0.238–0.302 ns, where the C32 satellite exhibited a smaller RMS of 0.196 ns compared to the other satellites of the same type. For SECM Satellites (C25–C30, C34, C35), due to fewer observations, C35 had a larger fitting RMS of 0.394 ns; C28 satellite observations were more than other satellites of the same type, and the minimum fitting RMS of 0.153 ns was achieved; the remaining SECM satellites had a fitted RMS of

0.171–0.269 ns. Overall, the clock offset fitting RMS of the SECM satellite was smaller than the result of the CAST satellite. Compared with the results of MGEX PCOs (S1), the estimated PCOs-only (S2) were used to reduce the RMS of the clock offset by 0.01 ns (3.9%), and the estimated PCOs and PVs (S3) were used to reduce fitting RMS by about 0.016 ns (6.5%).

## 6. Conclusions

When estimating the phase center offsets (PCOs) in-orbit, the x-offset parameter may have systematic variation within the β-angle. In this case, the average of the longer-term data was generally used as the final estimation of the x-offset. Considering that the period of the β-angle is half a year for the MEO satellites, at least one year of data is usually required. The systematic variation of x-offset with the β-angle is mainly due to the correlation between the solar radiation pressure (SRP) model. The absolute value of the β-angle is larger, the correlation is stronger. To solve this problem, a priori SRP model is added to assist the ECOM1, and apply a suitable $D_0$ constraint to reduce the correlation. Therefore, a stationary sequence of the x-offset estimations that were weakly related to the β-angle was obtained, and the final estimation of the x-offset can be obtained for a relatively short period, rather than a half year.

In this work, the average of the daily x-offset estimation of the China Academy of Space Technology (CAST) satellite was −0.321 ± 0.092 m, with a systematic variation of about 0.4 m by the ECOM1. After adding the a priori SRP model [15], a constraint of 1 nm/s$^2$ was also applied to the $D_0$ parameter, and a stationary sequence with the average of −0.227 ± 0.044 m was obtained. Comparing the two averages and STDs, the average of x-offset estimations had a difference of about 0.1 m, and the stability was improved by 50%. For the y-offset parameter, whether or not the a priori SRP model was added had no significant effect on the final estimation.

The estimated PCOs and PVs were compared with the Multi-GNSS Experiment (MGEX) PCOs from the precise orbit and clock offset. The Root Mean Square (RMS) of the 24 h orbit overlap with estimated PCOs and PVs was decreased by 6.5 mm (10.54%), 1.8 mm (4.4%), and 1.1 mm (8.03%) in the along-track, cross-track, and radial directions, respectively; the RMS and standard deviation (STD) of the 24 h clock offset overlap were decreased by 0.024 ns (8.6%) and 0.020 ns (13.1%), respectively; the fitting RMS of the 72 h clock offset of the quadratic polynomial was reduced for about 0.016 ns (6.5%).

From the comparison results, the precision of the orbit and clock offset with the estimated PCOs and PVs was improved. However, the improvements were mainly determined by the difference between the two models. Comparing the estimated PCOs and the MGEX PCOs in the x-offset component, the differences were 24 and 16 mm for CAST and the Shanghai Engineering Center for Microsatellites (SECM) satellites, respectively; for the y-offset, the difference was less than 9 mm; for the z-offset, the differences were 270 and 402 mm for CAST and SECM satellites, respectively. Considering that the BeiDou-3 Geostationary Earth Orbit (GEO) satellite ground station has too few observation data, the later launch of the BeiDou-3 Inclined Geosynchronous Orbit (IGSO) satellite results in a shorter data period, which is not enough to provide high-precision PCC estimations. These two types of satellites are not covered in this study.

Comparing the estimated PCOs of the Galileo Full Operational Capability (FOC) in-orbit [5] with the ground-based accurate phase center correction (PCC) model released by the European GNSS Agency (GSA), the differences between the two models were less than 5 mm in the x-offset and y-offset parameters, and the difference in the z-offset is about 152–304 mm. The similar difference in the z-offset of the BeiDou-3 MEO and Galileo might be caused by the unmodeled errors, and result in a scale error with the International Terrestrial Reference Frame 2014 (ITRF2014) [36]. This possible error might be the PCC model error at the receiver-end. Currently, IGS provides receiver antenna PCC correction for GPS and GLONASS frequencies only [4]. For BeiDou-2, BeiDou-3, and Galileo satellites, the GPS L1 and L2 model approximations are generally selected. Therefore, after the receiver antenna PCC calibrations of the Galileo and BeiDou frequencies are published, more accurate Galileo and BeiDou satellites antenna PCC parameters can be estimated in orbit.

**Author Contributions:** X.Y. and G.H. provided the initial idea for this study. X.Y. and L.W. designed the experiments. Z.Q. and S.X. performed the experiments. X.Y. analyzed the data and wrote the paper. G.H. and Q.Z. gave valuable advice on writing the paper.

**Funding:** This work was supported by the National Key R&D Program of China (2018YFC1505102), the Programs of the National Natural Science Foundation of China (41774025, 41731066), the Special Fund for Technological Innovation Guidance of Shaanxi Province (2018XNCGG05), the Special Fund for Basic Scientific Research of Central Colleges (grant no. CHD300102269305, CHD300102268305, Chang'an University), and the Grand Projects of the Beidou-2 System (GFZX0301040308).

**Acknowledgments:** The IGS MGEX, iGMAS are greatly acknowledged for providing the Multi-GNSS tracking data. Finally, the authors are also grateful for the comments and remarks of the reviewers, who helped to improve the manuscript significantly.

**Conflicts of Interest:** The authors declare no conflict of interest.

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
