# Peer review of "Estimation of the Antenna Phase Center Correction Model for the BeiDou-3 MEO Satellites"

_remotesensing, doi:10.3390/rs11232850_

Round 1

Reviewer 1 Report

Dear Authors,

I read your manuscript very carefully and I founded it detailed. The satellite antenna phase center offsets (PCOs) and phase variations (PVs) are used by GNSS Analysis Centers (i.e. IGS) for processing GNSS data in all its products and services. Thus, the estimation of the Antenna Phase Center Correction Model for the BeiDou-3 satellites, can provide a more accurate products and services, using multi-GNSS.   

Author Response

Thank you very much for your comments.

Reviewer 2 Report

The paper deals with a very useful and a necessary subject, namely solving of phase center corrections (PCC) of BeiDou 3 satellite antennas, which currently are not available and are required for precise BeiDou-3 PODs (precise orbit determinations) and navigation. However, there are some potentially misleading misconceptions, discussed below in details, namely that a POD iteration is necessary in order to determined PVs (phase center variations) and PCOs (phase center offsets), this needs to be corrected. Also, the English needs significant improvements, no attempts were made here to improve the English as there are too many places where it needs to be corrected/improved.

l .18 higher correlation between z-offset and the PVs ???

The correlation is so high that z PCO and PVs cannot be solved for simultaneously, either PV's or z PCO has to be fixed, (so that PV's is wrt the PCO), no iteration should help and be used, any z-PCO changes are likely due to solution uncertainties due to large PV's variations

22-26: For a reference, it would be useful to list here the magnitude of the original MGEX orbit overlap and clock fit RMS's, i.e. before the improvements. 37 : IERS Conventions (please spell out IERS) 38 : tracking station coordinates 48: "z-offset for IGSO, and satellites" ??? MEO ? 52: the MGEX Beidou-3 PV's are set to zeros?

There are many instances the English needs improvements so it can be understood what was meant, there are too many to be listed here, so from now on, only clear inaccuracies or misconceptions are noted, rather than the English improvements or grammatical corrections needed

71-72 : in the reference [19], GPS satellite (block specific PCV's were determined, fixing the PCO and subsequently the PCO's and PV were adjusted ( externally from the "raw" PV, removing the cosine trend, so no iterations!), whereas in [20], the PVs were fixed and z-PCO'z were adjusted to in a global adjustment to be consistent with the new ITRF scale, again, no iterations, please correct 75: GNSS data ! .. "and iteratively estimate the z-offset and PVs" iteratively??? 80: both PCO's and PV cannot be estimated within a global adjustment, either PCO's must be fixed (much like in [19] or PV fixed (like in [20], but not both!

Eq. 2: please define "delta z"

102: note that a separate adjustment of Eq. 2 is not an iteration! 105: the parameter 'a' is necessary mainly due to the condition of Eq. 3, not due to systematic errors

Eq. 4: is not a datum condition, it is a minimizing LS criterion of the separate adjustment of Eq. 2, where the LS residuals are PV's and the PV's /z-PCO datum is PVs=0! Please correct

122: for accelerations in the D and B directions, a_subD and a_subB

Tab 2: ISB is constant per each receiver ? Is this true and correct for each station during the whole period? Some stations may show significant variations from epoch to epoch, much like for Beidou-2. In general ISB should be treated as white noise, much like the clock solutions, to be on a safe side

Also, how were the eclipsing satellites treated?, deleted? This needs to be mentioned here

142: others, yellow points are MGEX stations of which the orange points can ... 144 Duplicate entries for C01 and C02 are to support the use of the legacy RINEX 3.01, 3.02 observation codes of B1I. and the code of the subsequent RINEX versions, respectively. 155 and in Figs 2, 3 legends : ... C21 satellite ... 157:... Galileo satellites..

Tab, 4 The MGEX PCO results for A and R orbit overlaps are better, why:?

207-212 The iteration, as already mentioned previously is unnecessary and even misleading. Any reasonable z-PCO (e.g. the MGEX one) can be used to solve (exactly) for PVraw, subsequently PV is determined by removing the cos variation and the constant according to Eq. 4, which yields PV and the correction to z-PCO. this was used by IGS (and in [4]), no iteration of POD is necessary!

Applies also to Fig. 6, the Fig. 6 (iteration 1) z-PCO values are in fact a priori values fixed in the iteration 1, so it should not be shown asn iteration 1 ) MGex ones could have been used), which is confusing and incorrect

The final z-PCO could have readily been obtained from the iteration 1 PVraw and an adjustment according Eq. 4 (no iteration wrt PV or z-PCO)

Fig. 7 Any explanations/reason why MGEX tracking of some BDS-3 satellites is so poor while iGMAS is OK? This is an important information for MGEX and BDS-3!

Tab 12 legend : ... quadratic polynomial fitting ..

l 333-336 should be deleted as the iteration is not necessary as discussed above

358-360 The vast majority of IGS receiver antenna are calibrated, I guess what is meant that they should also be calibrated for the Galileo and BDS frequencies ...

What about PCC for Beidou-3 GEO and IGEO satellites?

Reviewer 3 Report

Statement to the paper entitled „ Estimation of the Antenna Phase Center Correction Model for the BeiDou-3 MEO Satellites“.

Thank you for your paper, I enjoyed reading it. I think that new-generation BeiDou-3 satellites and their Antenna Phase Center Correction Model are very important, interesting and a topical issue at the moment.

Generally, the paper is high quality, nonetheless I have some minor comments and remarks.

1.

Page 2, line 47:

“However, there were significant differences between the estimations of the z-offset for IGSO, and satellites that prevented incorporation of these results into igs08.atx [9]”

What the authors mean as “significant differences”? – please add some explanation to the text.

Page 2, line 73:

“This study aims to estimate high accurate PCOs and PVs for BeiDou-3 MEO satellites.”

As previously: what the authors mean as “high accurate PCOs and PVs”? – please add some explanation to the text.

Page 4, Table 1:

It would be advisable to add more details about two listed a priori box-wing models (CAST and SECM), e.g. reasons for the differences in a1, a2, a3, a4, a5, a6, a7 coefficients.

Page 14, line 299-302:

“The RMS and STD of the C35 satellite were 0.34 ns and 0.25 ns, which showed a poorer accuracy than other satellites. One possible reason was lack of observations (seen from Figure 7), the another one was that the PCOs of C35 might be different from other satellites of SECM.”

A different PCOs can be a reason of poorer accuracy of clock offset? - it is unclear.

“After getting sufficient observations, further analysis of the specific reasons is required.”

In my opinion it could be check by calculating clock offset using the same (as for C35) number of data.

Page 16, line 354-356

“The similar difference in z-offset of the BeiDou-3 MEO and Galileo might be caused by the unmodeled errors and resulting in a scale error with the International Terrestrial Reference Frame 2014 (ITRF2014) [35]. The potential error might be the PCC model error at the receiver-end.”

…and any others unmodeled errors?

Page 16, line 358-360

“Therefore, after the PCC calibrations of the GNSS ground receivers are published, more accurate GNSS satellite PCC parameters can be estimated in orbit.”

What is the status of “PCC calibrations of the GNSS ground receivers”?

7 Some editorial corrections, e.g.:

Page 3, line 88 “[22]. the”

Page 13, figure 7 – the axis with BeiDou-3 satellites designations is difficult to read.
